# Seroprevalence of IgG Antibodies Directed against Dengue, Chikungunya and West Nile Viruses and Associated Risk Factors in Madagascar, 2011 to 2013

**DOI:** 10.3390/v15081707

**Published:** 2023-08-08

**Authors:** Anaïs Broban, Marie-Marie Olive, Michael Luciano Tantely, Anne-Claire Dorsemans, Fanjasoa Rakotomanana, Jean-Pierre Ravalohery, Christophe Rogier, Jean-Michel Heraud, Soa Fy Andriamandimby

**Affiliations:** 1Virology Unit, Institut Pasteur de Madagascar, Antananarivo 101, Madagascar; anais.broban@epicentre.msf.org (A.B.); marie-marie.olive@cirad.fr (M.-M.O.); anneclaire_dorsemans@yahoo.fr (A.-C.D.); jpierre@pasteur.mg (J.-P.R.); soafy@pasteur.mg (S.F.A.); 2Epicentre, 75017 Paris, France; 3Medical Entomoly Unit, Institut Pasteur de Madagascar, Antananarivo 101, Madagascar; lucinambi@pasteur.mg; 4Epidemiology and Clinical Research Unit, Institut Pasteur de Madagascar, Antananarivo 101, Madagascar; fanja@pasteur.mg; 5Directorate, Institut Pasteur de Madagascar, Antananarivo 101, Madagascar; christophe.rogier@primumvitare.com; 6Primum Vitare ! D’abord Prévenir !, 75015 Paris, France

**Keywords:** arbovirus, dengue virus, chikungunya virus, West Nile virus, serosurvey, risk factors, Madagascar, Africa

## Abstract

Arboviruses have been shown to circulate in Madagascar, including West Nile, dengue, and chikungunya viruses, though the extent of their circulation remains poorly documented. We estimated the seroprevalence of these three arboviruses in Madagascar and determined risk factors associated with seropositivity. Serum samples obtained from 1680 individuals surrounding the Sentinel Health Centers network in all regions of the country were analyzed using ELISA and hemagglutination inhibition assays for dengue, chikungunya, and West Nile viruses IgG antibodies, and multivariate logistic regression models were run. Overall, 6.5% [IC 95% 3.2–9.9] were seropositive for dengue virus, predominantly of Dengue serotype 1, 13.7% [IC 95% 6.5–20.9] for chikungunya virus, and 12.7% [IC 95% 9.0–16.5] for West Nile virus. There was no association with age, showing that dengue and chikungunya viruses were likely recently introduced. Eastern and Northern parts were more affected by dengue and chikungunya viruses, while West Nile virus seemed to circulate in all parts of the country. Dengue and chikungunya seropositivity were notably associated with high levels of vegetation, as well as frequent work in the forest, and West Nile seropositivity with the presence of cultivated areas, as well as standard of living. This analysis gives a new insight into arboviruses circulation and transmission patterns in Madagascar.

## 1. Introduction

Several arboviruses have been shown to circulate in Madagascar for a long time, including the West Nile virus (WNV), since the 1980’s [1]. Others, such as dengue (DENV) and chikungunya (CHIKV) viruses, were more recently documented [2].

DENV belongs to the *Flaviviridae* family. Dengue infection is currently considered a re-emergent disease, with an estimated 50–390 million cases occurring each year [3,4], causing a usually mild disease, though about 1% of cases turn hemorrhagic. Four main serotypes of the virus are identified (DENV1-4).

WNV also belongs to the *Flaviviridae* family. Wild and domestic birds are reservoirs of the virus, and humans are considered dead-end hosts. A majority of WNV infections are asymptomatic or mild, but it is estimated that less than 1% of infected patients develop a neuroinvasive form of the disease leading to encephalitis [5].

CHIKV belongs to the alphavirus genus of the *Togaviridae* family. The largest epidemic known worldwide is the one that took place in 2004–2011 in countries bordering the Indian Ocean [6]. It can easily be confused with dengue, exhibiting similar symptoms, although for chikungunya disease, arthralgia can persist for weeks or months [7].

DENV and CHIKV have been shown to co-circulate in many countries, including Madagascar [8]. While they are known to be transmitted by mosquitoes of the *Aedes* (*Ae.*) (*Stegomyia*) subgenus (mainly *Ae. albopictus* and *Ae. aegypti*), their circulation patterns remain poorly documented in the country. Until the end of the 1980s, none was detected through serological studies in Madagascar nor isolated [1]. In 2006, DENV1 and CHIKV were both detected in Toamasina (the East Coast of Madagascar) during an epidemic of febrile illness [2] and were shown to circulate in different regions along the East Coast [9]. Since 2007, the Institut Pasteur de Madagascar (IPM), in collaboration with the Malagasy Ministry of Health, has been coordinating a Sentinel Health Centers (SHC) Network that monitors epidemic-prone diseases (malaria, influenza, and arboviruses). Through this SHC, dengue-like syndromes accounted for 12.3% of fevers in 2008 [10]. The SHC and the National Reference Laboratory for Arboviruses located at IPM detected active dengue circulations in the cities of Toamasina, Antsiranana (North Coast of Madagascar), and Nosy Be (island of the west of Madagascar) in 2012 and more recently in 2020, and several chikungunya outbreaks in Toamasina (2010), Mananjary and Farafangana (2012, East coast), and Antsiranana (2013).

WNV has shown a different pattern in Madagascar, with mosquitoes from the *Culex* genus mainly involved in WNV transmission. Close to 30 mosquito species were identified as participating in WNV circulation in the country, and several bird species have been proven to act as reservoirs [11]. Local mosquito species (e.g., *Ae. Madagascarica, Anopheles (Cellia) pauliani* Grjebine) have also proved to be involved in transmission [12]. In 1990, a study showed high levels of WNV circulation in several regions of Madagascar [13], proving the virus had reached endemicity. Years later, another study confirmed the continued circulation of WNV in the Central Highlands and Western coast [14].

The objective of the current study was to estimate the seroprevalence of these three arboviruses in different areas of Madagascar through serum samples obtained in populations surrounding the SHC network in all regions of the country and analyzed for the presence of IgG antibodies.

## 2. Materials and Methods

### 2.1. Sample Collection

The study took place from 2011 to 2013 in SHC zones for surveillance of fever-associated diseases. The sampling protocol has been previously described [15]. In each of the 28 SHC zones, two sites were randomly selected, one near the SHC (usually urban or suburban) and another distant from SHC (within 5–15 km, usually rural) (Figure 1). Blood samples were obtained from thirty healthy adult individuals (≥18 years) in each of the 56 sites using spatial sampling.

### 2.2. Laboratory Analysis

All samples were first tested for IgG with in-house indirect Enzyme-Linked Immunosorbent Assay (ELISA) using antigens for ELISA produced by the Institut Pasteur de Laos. Briefly, each plate (96-Well Plates, 439454, Thermo Scientific™, Waltham, MA, USA) was coated with 100 µL of each antigen and incubated overnight at +4 °C. The samples were diluted 1:100. Peroxidase-labeled anti-human IgG (ref. 109-035-098–Jackson ImmunoResearch, West Grove, PA, USA) was used as conjugate. One positive and three negative control samples were included in each plate. ABTS (Ref 506200, KPL, Sylacauga, AL, USA) was used as substrate. Plate validation was determined according to Westgard rules [16]. Samples with ELISA DENV Optical Density (OD) > 0.08 or WNV OD > 0.02 were further analyzed with Hemagglutination Inhibition (HI) assays for DENV1-4 and West-Nile antigens (WNV). HI analyses were performed as previously described [17]. Antigens for HI were locally produced at the Institut Pasteur de Madagascar using the sucrose-acetone extraction method from mouse brains. Validation of antigens titers used HI techniques [17]. 

A specimen was considered positive for CHIKV when ELISA OD was equal to or greater than 0.4. A sample was considered positive for a DENV or WNV if its HI titer were equal or greater than 1:80. If HI titers were equal or greater than 1:80 for all DENV1-4 serotypes and WNV, virus(es) with the weakest titer were considered negative.

### 2.3. Environmental Data Collection

To preserve the maximum spatial specificity and represent the very local environment due to the limited movement capacity of mosquitoes, environmental data were obtained by clusters of households. A total of 216 household clusters (max 200 m) were identified, accounting for 1 to 24 individuals. Cluster coordinates were those of the centroid of all household coordinates constituting the cluster. Each location was characterized for each environmental variable using a 5-year period before the beginning of the study, from the 1 January 2007 to the 31 December 2011. 

Land Surface Temperature for Day and Night (LST Day and LST Night), Normalized Difference Vegetation Index (NDVI), and precipitations were obtained using satellite images downloaded from IRI Data Library. For each of the four variables, 4 images of each year were used to determine a mean value over 5 years, for a total of 20 images covering equally all the seasons. The resolution was 1 km for LST Day and Night, 50 km for precipitation, and 250 m for NDVI. 

Land cover variables were obtained using a 30 m resolution image from the Madagascar Vegetation mapping project (2007) [18]. A buffer zone of diameter 200 m was defined around each of the 216 cluster coordinates, and this surface was used to determine a percentage of the different land covers present in the direct environment of each cluster. Percentages were then recategorized into 5 variables (water and wetlands, cultivations, humid forest, grassland and dry forest, and bare soil/rock). Altitude data were extracted from the Shuttle Radar Topography Mission (SRTM), using the same diameter 200 m buffer zone. Extraction of the data was performed using QGIS software. 

### 2.4. Multiple Factors Analysis (MFA)

Synthetic variables characterizing the environment of clusters were computed using MFA analysis run in R software, as previously described [15]. By performing a factor analysis, the MFA summarizes the initial set of environmental variables in quantitative variables (so-called dimensions). 

### 2.5. Logistical Regression Analysis

Analysis was conducted using Stata13 software. Confidence intervals were adjusted for the clustering effect on the 56 sites. The age categories used were 18–24, 25–34, 35–44, and 45+. Variables with missing values of up to 2% were inferred. Logistic regression analysis was run using the xtlogit command, and variables with a *p*-value lower than 0.2 were selected for multivariate analysis. Models were adjusted using descending method until all remaining variables appeared significant, and the AIC indicator was compared to select final models.

Variables tested included personal variables (age, sex, education level, smoker, outdoor profession), general environment variables (distance to SHC, rural/urban, MFA factors), individual habits (frequent activities in rice fields and forest, frequent use of bed nets, frequent contacts to water bodies), house variables (electricity and running water in the house, smoke in the house when cooking, type of construction materials, house targeted for Indoor Residual Spraying (IRS) in the last 12 months), and house environment variables (biological and non-biological wastes around the house, cultures and water within 10 m around the house).

## 3. Results

Overall, we enrolled 1680 individuals aged 18 years and more. Half of them (50.7%) were female, and the mean age was 37.7 years (ranging from 18 to 99 years). 

### 3.1. Seroprevalence Results

#### 3.1.1. Dengue Virus

There was an overall DENV seroprevalence of 6.5% [IC 95% 3.2–9.9], with regional disparities. Among the 28 zones, the highest seroprevalences were observed in Nosy Be (North, 48.3%) and Toamasina (East Coast, 43.3%) (Figure 1, Table 1). Univariate logistic regression test showed no association with age (*p* = 1.000), sex (*p* = 0.4188), or distance to SHC (*p* = 0.2168) (see Appendix A for complete univariate results). Among the 110 positive individuals for at least one dengue serotype, 66 (60.0%) were seropositive for one serotype only, and others tested positive for several serotypes, for a total of 186 infections. Among the seropositive participants, 85.5% [IC 95% 78.2–92.7] tested positive for DENV1, 30.0% [IC 95% 19.3–40.7] for DENV2, 40.0% [IC 95% 27.9–52.1] for DENV3, and 13.6% [IC 95% 6.3–21.0] for DENV4. Spatial distribution was globally similar for all serotypes (Appendix A).

#### 3.1.2. Chikungunya Virus

Overall IgG CHIKV seroprevalence was 13.7% [IC 95% 6.5–20.9], ranging from 0% to 86.7% in Nosy Be. Most locations with high seropositivity levels were concentrated on the East Coast and north of the country (Figure 1 and Table 1). Univariate logistic regression showed no association with age (*p* = 0.8143), distance to SHC (*p* = 0.1767), or sex (*p* = 0.9679) (see Appendix A for complete univariate results).

#### 3.1.3. West Nile Virus

WNV seroprevalence was estimated to be 12.7% [IC 95% 9.0–16.5], ranging from 0% in Ambositra and Anjozorobe to 45.0% in Toamasina. Locations with high seropositivity levels were distributed across the whole country (Figure 1 and Table 1). Univariate logistic regression showed no association with age (*p* = 0.2498) or sex (*p* = 0.7432). Nevertheless, seroprevalence was significantly higher in sites distant from the SHC (17.0% [IC 95% 12.8–21.2], *p* = 0.0005) (see Appendix A for complete univariate results).

### 3.2. Logistical Regression Results

#### 3.2.1. Multiple Factor Analysis for Environmental Variables

Four MFA factors contributing to 72% of the total variance were selected. Table 2 shows the correlation between each of these four factors and quantitative covariates included in the MFA (LSTN, LSTD, NDVI, Precipitations), as well as the estimation of the association with qualitative covariates (land cover variables, category Yes).

In these results: Factor 1 positivity represents ecosystems with high day and night temperatures, with low precipitation and NDVI, associated with the absence of wooded vegetation.Factor 2 positivity represents ecosystems with high NDVI and relatively high night temperatures, without grassland and moderate levels of cultivated areas, but with the important presence of humid forests and wetlands.Factor 3 positivity represents ecosystems with high day and night temperatures, with moderate precipitation dominated by the presence of cultivated areas but no bare soil or water bodies.Factor 4 positivity represents areas with moderate temperatures and NDVI but with a low surface of water, moderate levels of forests, and the important presence of bare soil.

#### 3.2.2. Multivariate Regression Models

Multivariate logistic regression models selected are presented in Table 3 below. All three viruses were associated with factor 2, characterized by high NDVI and the presence of humid forests and wetlands. Dengue seropositivity was also associated with factor 4, described by moderate precipitations and forest landscapes and the important presence of bare soil. On the other hand, WNV seropositivity was associated with areas predominantly covered by cultivated zones (Factor 3).

Frequent work in forests appeared as a risk factor, and frequent activities in rice fields as a protective factor for both dengue and chikungunya seropositivity. Moreover, living in a house targeted for an anti-mosquito spraying program in the last 12 months appeared protective for testing positive for antibodies for these two viruses. Smoking was associated with less seropositivity for dengue.

The risk factor analysis also shows that chikungunya positivity is associated with urban settings, with rural and semi-urban settings appearing as protective and non-biological wastes around the house as a risk. This association is not apparent with dengue seropositivity.

West Nile virus seropositivity was, on the contrary, associated with activities in contact with water bodies. Electricity in the house, here as a proxy of level of life, appeared as a protective factor for this virus seropositivity, though most households did not have a sufficient standard of living to access it at the time of the survey.

## 4. Discussion

The above results confirm that all three viruses circulated in Madagascar before or during the study period. 

### 4.1. Dengue and Chikungunya Viruses

DENV and CHIKV results seem consistent with previous reports in the country [9], including during the 2006 epidemic in the Toamasina region [2]. 

When looking at the national circulation of these viruses, it appears that they mainly circulate on the East Coast and the northern part of the country and, to a lesser extent, in the Central Highlands regions. The multivariate risk factor analysis, which highlights the importance of high NDVI temperatures and the presence of humid forests and wetlands, is coherent with this observation and bio-climatic mapping of the country [19]; moreover, these have been shown to influence dengue transmission in other contexts [20]. 

Interestingly, we observed positive individuals among those living in the Central Highlands of Madagascar. Although no epidemic has been reported from this region, there is evidence of *Aedes (Stegomyia)* presence [21,22]. The notion of travel was requested in the individual questionnaire, but they were only being asked to report travel during the last previous years; thus, we cannot exclude that those positive individuals had traveled some years ago. 

No significant association was found with age, and seropositivity affected all age groups equally, confirming that transmission of these two viruses in Madagascar is likely contemporary to our study [2]. However, it does not allow us to conclude whether the diseases became endemic ever since; it would require new seroprevalence assessments, including younger age categories born after the most recent known epidemic (2006-20007). Moreover, as a previous study in an African context identified young age as a risk factor for dengue exposure occurrence [23], this work strongly supports the need for further investigation to explore the relation between age and exposure.

For both dengue and chikungunya antibodies, frequent contact with forest appeared as a risk factor. This is probably linked to the living habits of community members in the east and northern parts of the country, who conduct agricultural labor during the day for a long period (five months) to harvest and process coffee, cacao, and vanilla crops, thus being exposed to diurnal species such as *Aedes aegypti* and *Aedes albopictus* [1]. Indeed, both species are known to grow in coffee tree holes, bamboo cuts, and *Pandanus* plants leaf axils present in that plantation [24]. Moreover, *Aedes (Stegomyia)* is known to be more abundant in partial or full-shaded locations (forest) [25], while sunlight and high temperatures of the rice fields should reduce the flight and biting activities of mosquito vectors. 

Similarly, in our analysis, living in a house targeted for Indoor Residual Spraying (IRS) in the last 12 months appeared as a protective factor for seropositivity to both viruses. Although not documented in Madagascar, the effectiveness of IRS as a dengue control intervention by reducing the adult stage of *Aedes* mosquito populations, or reducing endophilic and endophagic *Stegomyia* species populations linked to the presence of insecticide-treated surfaces, was already reported in other countries [26,27]. Nevertheless, considering that the antibody protection we measured is long-term, this result should be interpreted with caution. Another hypothesis could be that targeted households are the ones that regularly participate in this kind of program, thus accessing more protection material distributions and sensitization sessions about protecting individuals against mosquitoes, or that presence of impregnated mosquito nets in the house, distributed at the time of spraying, helps to keep mosquitoes away.

Interestingly, smokers also appeared to be protected against dengue in our multivariate model. Moreover, it has been reported that lower endophilic mosquito density was observed in the habitations of tobacco smokers compared to the habitation of non-smokers [28]. Few observational studies have been conducted in this area. Nevertheless, the role of nicotine on insect resistance in some plants may be a possible explanation [29], needing more investigation. 

Living in urban areas and with non-biological wastes displayed around the house has been found to be associated with seropositivity of chikungunya disease. This is consistent with a previous study in Madagascar, showing that *Ae. albopictus* vector is widely present in the country. Its breeding sites are mostly made of artificial environments such as dumped containers, used and abandoned tires and buckets, coconuts, and bamboo-cut trees. Those are more likely to be present in cities and inhabited places [22]. Nevertheless, these risk factors do not appear significant for dengue and remain to be further studied.

Though living in urban areas is usually considered a risk factor for dengue disease as well [4], higher circulation in rural areas was previously reported in some contexts [30]. In Madagascar, our analysis shows it is not associated with urban settings or the presence of wastes; this may suggest slight differences in transmission patterns compared to chikungunya, and this element should be further explored to better understand the DENV transmission pattern in the country. Differences in transient dynamics and long-term endemic levels between chikungunya and dengue, with a risk of invasion or an outbreak varying with vector-virus assemblages [31], should not be excluded.

Finally, this study is the first national serological survey that assesses the spread of DENV and CHIKV in Madagascar as well as the different DENV serotypes that circulate in the country. Our data showed that the DENV1 serotype seemed to be the predominant circulating strain, though we demonstrated reactivities with all serotypes with different intensity levels (Appendix A). DENV1 was responsible for an outbreak in Toamasina in 2006 [2]. Recent surveillance reports have confirmed the presence of DENV1, DENV2, and DENV3 in the country [32], being consistent with these findings. As DENV4 has never been confirmed in the country, and while our results show lower levels of seropositivity to this serotype, the risk of serotype cross-reactivity has to be considered, and its circulation should be assessed in further studies before formal conclusions.

### 4.2. West Nile Virus 

Our study is consistent with previous reports of WNV endemicity in the country. The virus seems to circulate in all regions, including Central Highlands, though to a lesser extent. Though the multivariate analysis results showed that seropositivity is associated with certain types of landscapes (high NDVI, relatively high temperatures, and cultivated areas, or humid forest and wetland), it is believed that WNV circulation is mostly driven by vector and reservoirs presence, with many mosquito species listed as potentially associated with transmission in Madagascar [11]. 

Despite endemicity, IgG seroprevalence levels do not appear to increase compared to previous studies [13,14], and no association with age was reported. This could be explained by the long-term waning of WNV IgG antibodies [33]; it suggests that no significant increase in WNV circulation had happened in the previous years, and the level of WNV transmission is stable in the country. 

WNV transmission was significantly more important in sites distant from SHC, which are less urban and where contact with birds and other animals is expected to be more frequent. This finding is also coherent with the results of multivariate analysis, showing WNV seropositivity is associated with landscapes dominated by forests and cultivated areas, as well as frequent activities in relation to water bodies (fishing, swimming). This last association can be put in relation to the presence of vector mosquitoes close to water bodies [11,12] and underlines the importance of reservoirs in WNV circulation in Madagascar. Generalist feeder species (such as *Culex. antennatus*, *Cx univittatus*, and *Cx quinquefasciatus*) acting as bridge vectors between wild waterbirds, domestic village birds, and humans were also found in abundance in rural areas [34].

Electricity in the house can be considered as a proxy for a higher standard of living, though imperfect when whole rural villages could be deprived of electricity. Living in a household with electricity was associated with lower WNV seropositivity, showing that the populations most affected were those living in the poorest or rural areas. Though it is not a commonly reported risk factor as such, previous studies have reported a poor level of education as a potential risk factor for WNV infection in Nigeria, which could be consistent with our results [35]. In the USA, the population in poverty is highly exposed to WNV incidence, probably due to the quality of housing, which facilitates the entrance of mosquito vectors into the house [36].

Despite analysis considering the local environment and variables, it is noted that different mosquitoes involved in West Nile transmission could have other behaviors that are not acknowledged in this model.

### 4.3. Limitations 

Our study has some limitations. First, while a non-negligible part of dengue seropositive participants was positive to several serotypes, it is known that cross-reactions between different DENV serotypes are frequent, as well as with some other flaviviruses. Therefore, we cannot rule out this aspect that may overestimate the seroprevalence of some serotypes or of WNV.

Moreover, although unlikely, we cannot exclude that some seropositive individuals acquired their infection in other regions of the country or even internationally on very rare occasions. 

Finally, the structure of our data with small samples stratified over many places with different climate and mosquito circulation profiles may have led to confusion bias for some variables (including the use of bed nets) (Appendix A).

## 5. Conclusions

The national extended circulation of arboviruses, especially DENV and CHIKV, should be considered by health authorities and workers during an event of acute febrile illness outbreaks in the community. Indeed, these viruses are only considered in coastal areas where previous epidemics have been reported, while our analysis identifies favorable transmission settings. 

DENV seemed to be best transmitted in areas with high NDVI and temperatures, characterized by humid forests, wetlands, or bare soil, and favored by forest activities. Chikungunya transmission also seemed to be happening in similar conditions, though our results show that it is also favored by living in urban settings and the presence of non-biological wastes around the house. Finally, West Nile transmission seems to happen in most climate areas of the country and is linked to contact with water bodies and poor levels of life.

Although our data are in favor of a relatively recent introduction of DENV and CHIKV viruses in Madagascar (prior to 2011), more recent serosurveys should be conducted to update the epidemiologic situation and investigate potential evolutions to endemicity, especially including younger ages. Our results also show continued endemicity of WNV throughout the country. Risk factors yielded by multivariate analysis are globally coherent results with known transmission patterns of the viruses, though specific Madagascar context, population activities and habits, and mosquito presence patterns seem to also play a role. The effects of bioclimates and other risk factors should also be further studied to better understand the transmission patterns of arboviruses in Madagascar.

## Figures and Tables

**Figure 1 viruses-15-01707-f001:**
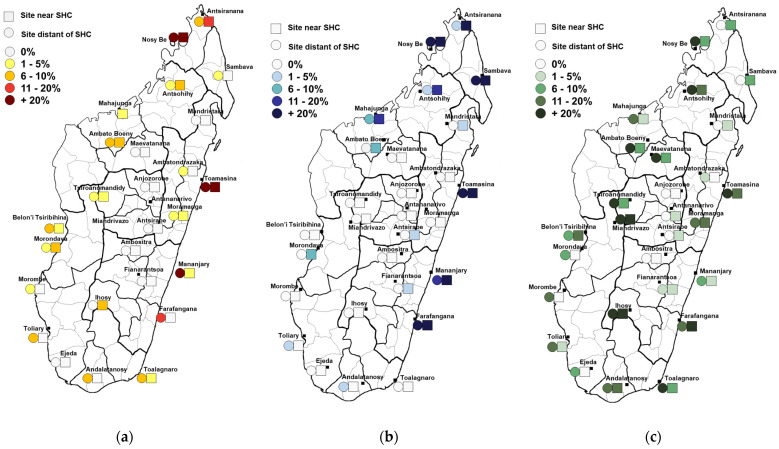
Spatial distribution of seropositivity for the different arboviruses tested. (**a**) Dengue virus seropositivity. Participants are considered positive for dengue virus if positive for at least one serotype. (**b**) Chikungunya virus seropositivity. (**c**) West Nile virus seropositivity.

**Table 1 viruses-15-01707-t001:** Detailed seropositivity statistics for dengue, chikungunya, and West Nile viruses according to study sites. Participants are considered positive for dengue if positive for at least one serotype. Sixty participants were enrolled around each Sentinel Health Center (SHC), thirty were from near the center (urban), and thirty were distant from the center (rural).

City	Dengue Virus			Chikungunya Virus			West Nile Virus		
	Nb Positive	(%)	CI *	Nb Positive	(%)	CI *	Nb Positive	(%)	CI *
Ambato Boeny	4	6.7	1.9–11.4	1	1.7	0.0–4.0	10	16.7	7.1–26.2
Ambatondrazaka	1	1.7	0.0–4.0	0	0.0	0.0–0.1	1	1.7	0.0–4.0
Ambositra	0	0.0	0.0–0.1	0	0.0	0.0–0.1	0	0.0	0.0–0.1
Andalatanosy	2	3.3	0.0–8.1	1	1.7	0.0–4.0	11	18.3	16.0–20.7
Anjozorobe	0	0.0	0.0–0.1	0	0.0	0.0–0.1	0	0.0	0.0–0.1
Antsohihy	4	6.7	1.9–11.4	7	11.7	0.0–23.6	18	30.0	6.2–53.8
Antananarivo	1	1.7	0.0–4.0	0	0.0	0.0–0.1	1	1.7	0.0–4.0
Antsirabe	0	0.0	0.0–0.1	1	1.7	0.0–4.0	1	1.7	0.0–4.0
Antsiranana	7	11.7	4.5–18.8	8	13.3	0.0–27.6	3	5.0	0.0–12.1
Belon’i Tsiribihina	3	5.0	0.0–12.1	0	0.0	0.0–0.1	6	10.0	5.2–14.8
Ejeda	0	0.0	0.0–0.1	0	0.0	0.0–0.1	2	3.3	0.0–8.1
Farafangana	4	6.7	0.0–16.2	37	61.7	59.3–64.0	15	25.0	17.9–32.1
Fianarantsoa	0	0.0	0.0–0.1	1	1.7	0.0–4.0	2	3.3	0.0–8.1
Ihosy	3	5.0	0.0–12.1	0	0.0	0.0–0.1	20	33.3	23.8–42.9
Maevatanana	0	0.0	0.0–0.1	0	0.0	0.0–0.1	11	18.3	6.4–30.2
Mahajanga	1	1.7	0.0–4.0	8	13.3	3.8–22.9	6	10.0	5.2–14.8
Mananjary	9	15.0	0.0–31.7	22	36.7	8.1–65.3	4	6.7	1.9–11.4
Mandritsara	0	0.0	0.0–0.1	1	1.7	0.0–4.0	2	3.3	0.0–8.1
Miandrivazo	1	1.7	0.0–4.0	0	0.0	0.0–0.1	15	25.0	22.6–27.4
Moramanga	2	3.3	0.0–8.1	0	0.0	0.0–0.1	9	15.0	12.6–17.4
Morombe	1	1.7	0.0–4.0	0	0.0	0.0–0.1	5	8.3	0.0–20.2
Morondava	3	5.0	2.6–7.4	2	3.3	0.0–8.1	3	5.0	0.0–12.1
Nosy Be	29	48.3	26.9–69.8	52	86.7	81.9–91.4	14	23.3	4.2–42.4
Sambava	1	1.7	0.0–4.0	46	76.7	52.8–100.0	2	3.3	0.0–8.1
Taolagnaro	4	6.7	1.9–11.4	0	0.0	0.0–0.1	11	18.3	1.6–35.0
Toamasina	26	43.3	30.4–56.2	42	70.0	0.0–0.1	27	45.0	0.5–19.5
Toliary	2	3.3	0.0–8.1	1	1.7	0.0–4.0	6	10.0	5.2–14.8
Tsiroanomandidy	2	3.3	0.0–8.1	0	0.0	0.0–0.1	9	15.0	3.1–26.9
**TOTAL**	**110**	**6.5**	**3.2–9.9**	**230**	**13.7**	**6.5–20.9**	**214**	**12.7**	**9.0–16.5**

* Adjusted for clustering effect on the 56 sites. In sites where no samples tested positive, one-sided 97.5% confidence intervals were estimated using an exact unadjusted method.

**Table 2 viruses-15-01707-t002:** Correlation between quantitative and qualitative covariates included in the MFA and each selected factor.

Covariates	Factor 1	Factor 2	Factor 3	Factor 4
Land Surface Temperature Night (LSTN)	0.6359734	0.4834523	0.3790634	0.1336330
Land Surface Temperature Day (LSTD)	0.6743499	/	0.5535304	/
Normalized Difference Vegetation Index (NDVI)	−0.6806744	0.4492561	/	0.1418775
Precipitations	−0.2008070	−0.3733040	0.2891146	0.2345488
Land Cover Grassland	/	−1.2587948	/	/
Land Cover Water/wetlands	0.6192618	0.8190209	−0.4854862	−0.7919986
Land Cover Bare soil	0.9902471	0.4585246	−0.5887855	0.8189414
Land Cover Cultivated areas	−0.3448483	0.3614344	0.6748149	−0.1676591
Land cover Humid Forest	−1.3089895	0.9065925	/	0.2996699

/: The correlation coefficients were not significantly different from zero and so were not included in the results.

**Table 3 viruses-15-01707-t003:** Multivariate analysis results for dengue, chikungunya, and West Nile viruses seropositivity.

Virus	Tested Variable	N Tested ^1^	Nb Pos (%)	IC Pos	Adjusted OR	IC 95% OR	*p*-Value
Dengue	MFA Factor 2	1678			1.66	1.24–2.18	0.001
	MFA Factor 4	1678			1.88	1.30–2.72	0.001
	Smoker						0.031
	No	1334	93 (7.0)	4.8–9.1	1		
	Yes	344	17 (4.9)	2.5–7.4	0.48	0.26–0.89	
	Frequent work in rice fields						<0.001
	No	909	77 (8.5)	5.5–11.4	1		
	Yes	769	33 (4.3)	2.6–6.0	0.36	0.21–0.63	
	Frequent activities in the forest						0.048
	No	1083	58 (5.4)	3.4–7.3	1		
	Yes	595	52 (8.7)	5.4–12.1	1.72	1.00–2.94	
	House targeted for anti-mosquito spraying program (last 12 months)						0.002
	No	1190	103 (8.7)	6.1–11.3	1		
	Yes	488	7 (1.4)	0.3–2.6	0.24	0.10–0.59	
Chikungunya	MFA Factor 2	1678			2.48	1.37–4.46	0.003
	General environment						0.031
	Urban	516	105 (20.3)	13.2–27.5	1		0.008
	Peri-urban	450	21 (4.7)	1.4–7.9	0.14	0.04–0.51	0.003
	Rural	714	104 (14.6)	6.8–22.3	0.25	0.06–0.99	0.048
	Frequent work in rice fields						0.018
	No	911	162 (17.8)	12.8–22.7	1		
	Yes	769	68 (8.8)	3.6–14.1	0.42	0.21–0.86	
	Frequent activities in the forest						0.015
	No	1084	133 (12.2)	8.5–16.1	1		
	Yes	596	97 (16.3)	8.9–23.7	2.27	1.17–4.39	
	Non-biological wastes next to the house						0.041
	No	457	25 (5.5)	2.8–8.1	1		
	Yes	1221	205 (16.8)	11.6–21.9	2.04	1.03–4.03	
	House targeted for anti-mosquito spraying program (last 12 months)						<0.001
	No	1190	228 (19.2)	13.7–24.6	1		
	Yes	488	2 (0.01)	0.00–0.01	0.02	0.00–0.18	
West Nile	MFA Factor 2	1678			1.37	1.08–1.75	0.010
	MFA Factor 3	1678			1.48	1.11–1.97	0.007
	Frequent contacts to water bodies (fishing, swimming)						0.033
	No	1564	195 (12.5)	10.0–14.9	1		
	Yes	116	19 (16.4)	8.1–24.6	2.07	1.06–3.84	
	Electricity in the house						0.001
	No	1114	180 (16.2)	12.8–19.5	1		
	Yes	564	34 (6.0)	3.7–8.3	0.42	0.25–0.70	

^1^ Two participants were missing the full household questionnaire and were therefore excluded from this analysis.

## Data Availability

The data presented in this study are available on request from the corresponding and last authors. The data are not publicly available due to ethical considerations.

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
