# Peer review of "Seroprevalence of IgG Antibodies Directed against Dengue, Chikungunya and West Nile Viruses and Associated Risk Factors in Madagascar, 2011 to 2013"

_viruses, 2023, doi:10.3390/v15081707_

Round 1

Reviewer 1 Report

In this study the authors are reporting on the seropositivity of DENV, WNV and CHKV in Madagascar. The results presented here are very important especially in undiagnosed febrile cases and differential diagnostic testing of DENV, CHKV and WNV should probably be included although more recent data will be beneficial. Overall, the study was well planned with solid statistical analyses and interesting results. The English langue and writing could be improved.

Introduction:

Overall, the authors use telegram style writing and not real long sentences. The writing style needs to be adjusted.

Methods:

The authors are explaining the statistical analyses very well but is lacking in the laboratory analyses. The methods should be explained so that anyone reading this can duplicate the study. Confirming the results with live virus neutralizations would also strengthen the results especially with regards to the Flaviviruses. The authors could also test IgM to differentiate between new infections which is more indicative of an outbreak. Also see minor comments.

Minor comments:

Line 24: Replace “in” with “from”

Line 31: West nile virus

Line 61: Regions

Line 74: Give an example of the local mosquitoes

Line 75: “into” not the correct word

Please specify which ELISA plates were used. If they were commercially available, please add the supplier, country and catalogue number. If they were lab optimized, please add all methods used as well as consumable/reagents.

Please reference antigens production or provide more details eg optimization etc.

Rearrange section 2.2 as to 1st discuss ELISA for all 3 viruses and then HI results.

Line 234: Those results…rather say the above results or our study

Line 274: Delete “we”

Line 277: Mosquitoes

Line 280-282: Any idea why smokers seem to be protected? Maybe give a possible explanation.

Line 285-289: Long sentence with multiple use of “which” .

Line 298” Should no be excluded

The English langue and writing could be improved.

Author Response

Reviewers 1

In this study the authors are reporting on the seropositivity of DENV, WNV and CHKV in Madagascar. The results presented here are very important especially in undiagnosed febrile cases and differential diagnostic testing of DENV, CHKV and WNV should probably be included although more recent data will be beneficial. Overall, the study was well planned with solid statistical analyses and interesting results. The English langue and writing could be improved.

Authors’ response: Authors would like to thank the reviewer for the time spent to read and review our manuscript and for pertinent and constructive comments.

Introduction:

Overall, the authors use telegram style writing and not real long sentences. The writing style needs to be adjusted.

Authors’ response: Thank you for this commentary. We also tried to adjust some English writing elements accordingly throughout the manuscript, to avoid this inconvenience.

Methods:

The authors are explaining the statistical analyses very well but is lacking in the laboratory analyses. The methods should be explained so that anyone reading this can duplicate the study. Confirming the results with live virus neutralizations would also strengthen the results especially with regards to the Flaviviruses. The authors could also test IgM to differentiate between new infections which is more indicative of an outbreak. Also see minor comments.

Authors’ response: Thank you for this remark. More details were added in the text to explain laboratory analysis (ELISA).

Minor comments:

Line 24: Replace “in” with “from”; Line 31: West nile virus; Line 61: Regions

Authors’ response: changes have been made as requested by the reviewer.

Line 74: Give an example of the local mosquitoes.

Authors’ response: examples have been added in the text.

Line 75: “into” not the correct word

Authors’ response: changes have been made as requested by the reviewer.

Please specify which ELISA plates were used. If they were commercially available, please add the supplier, country and catalogue number. If they were lab optimized, please add all methods used as well as consumable/reagents.

Authors’ response: changes have been made as requested by the reviewer. More details on ELISA method were added in the text.

Please reference antigens production or provide more details eg optimization etc.

Authors’ response: More explanation was added in the text. Antigen preparation used techniques described by Clarke, D.H.; Casals, J cited previously.

Rearrange section 2.2 as to 1st discuss ELISA for all 3 viruses and then HI results.

Authors’ response: changes have been made as requested by the reviewer.

Line 234: Those results…rather say the above results or our study; Line 274: Delete “we”; Line 277: Mosquitoes

Authors’ response: changes have been made as requested by the reviewer.

Line 280-282: Any idea why smokers seem to be protected? Maybe give a possible explanation.

Authors’ response: a possible explanation might be the role of nicotine in insect resistance, however its real mechanism in human smoker needs to be more to be more investigated but it was added in the text.   

Line 285-289: Long sentence with multiple use of “which”; Line 298” Should no be excluded.

Authors’ response: changes have been made as requested by the reviewer.

Reviewer 2 Report

In this study, the authors estimated the prevalence of antibodies specific to dengue (DENV), chikungunya (CHIKV) and West Nile (WNV) viruses, in serum samples from 1680 adult individuals recruited between 2011 and 2013, in 28 Sentinel Health Centers zones distributed across the territory of Madagascar. They found that 6.5%, 13.7%, and 12.7% of the participants were seropositive for DENV, CHIKV and WNV, respectively. Most of the individuals seropositive for DENV and CHIKV were identified in Eastern and Northern parts, while those seropositive for WNV were found in all parts of the country. Results from the study suggested a recent introduction of both DENV and CHIKV, and endemic circulation of WNV.

Although the results provided by this study are somewhat dated (samples were collected during 2011-2013), they remain important from an epidemiological and public health perspective, in a country where data on the circulation of arboviruses is limited. The following corrections should be made before the paper can proceed to publication.

General comments:

- capital letters for "dengue" and "chikungunya" should be removed throughout the manuscript.

- dengue, chikungunya and West Nile viruses should be replaced by their respective acronyms in several parts of the manuscript.

Specific comments:

ABSTRACT:

- Line 28: "for chikungunya virus"

- Line 29: "for West Nile virus"

- Line 31: "West Nile virus seemed"

INTRODUCTION:

- Line 51: better use "genus" instead of "group"

- Line 55: remove "viruses"

- Line 56: "Aedes (Ae.)"

- Line 59: replace "DENV-1" by "DENV1"

- Line 60: remove "viruses"

- Line 61: "were shown"

- Line 61: "different regions"

MATERIALS AND METHODS:

- Line 92: please specify what kind of antibodies (IgG I suppose) were detected by ELISA.

- Line 96: "and WNV antigens."

- Line 101: remove "virus"

- Line 102: "were equal"

RESULTS:

- Line 151: Did you specifically detect IgG antibodies for DENV?

If you used the results of the HI assay to assess DENV seroprevalence (this is what I suppose based on the caption of Table 1 stating “Participants are considered positive for dengue if positive for at least one serotype”), I suggest removing "IgG".

DISCUSSION:

- Line 298: "should not be excluded"

- Line 299: "that assesses"

- Line 301: "DENV1 serotype seemed"

- Line 303: "was responsible for"

- Line 306: "DENV4 has never been confirmed"

- Line 309: "West Nile virus"

- Line 317: Did you specifically detect IgG antibodies for WNV?

- Line 347: "some other flaviviruses"

- Lines 349-350: some seropositive individuals could also have acquired their infection in another country.

CONCLUSIONS:

- Lines 356-357: "an event of acute febrile illness outbreak"

- Line 358: "previous epidemics"

- Line 360: "DENV seemed"

- Line 364: "WNV transmission seemed"

SUPPLEMENTARY MATERIALS:

Tables S1, S2 and S3: please indicate in the caption the signification of p-values shown in bold.

Table S1: There is an error for the % of positive individuals among those aged 18-24 years (it should be 6.7% instead of 26.7%).

Author Response

Reviewers 2

In this study, the authors estimated the prevalence of antibodies specific to dengue (DENV), chikungunya (CHIKV) and West Nile (WNV) viruses, in serum samples from 1680 adult individuals recruited between 2011 and 2013, in 28 Sentinel Health Centers zones distributed across the territory of Madagascar. They found that 6.5%, 13.7%, and 12.7% of the participants were seropositive for DENV, CHIKV and WNV, respectively. Most of the individuals seropositive for DENV and CHIKV were identified in Eastern and Northern parts, while those seropositive for WNV were found in all parts of the country. Results from the study suggested a recent introduction of both DENV and CHIKV, and endemic circulation of WNV.

Although the results provided by this study are somewhat dated (samples were collected during 2011-2013), they remain important from an epidemiological and public health perspective, in a country where data on the circulation of arboviruses is limited. The following corrections should be made before the paper can proceed to publication.

Authors’ response:  authors would like to thank the reviewer for the time spent to read and review our manuscript and for suggesting relevant ways to improve the entire manuscript.

General comments:

- capital letters for "dengue" and "chikungunya" should be removed throughout the manuscript.

- dengue, chikungunya and West Nile viruses should be replaced by their respective acronyms in several parts of the manuscript.

Authors’ response: changes have been made as requested by the reviewer.

Specific comments:

ABSTRACT:

- Line 28: "for chikungunya virus"

- Line 29: "for West Nile virus"

- Line 31: "West Nile virus seemed"

INTRODUCTION:

- Line 51: better use "genus" instead of "group"

- Line 55: remove "viruses"

- Line 56: "Aedes (Ae.)"

- Line 59: replace "DENV-1" by "DENV1"

- Line 60: remove "viruses"

- Line 61: "were shown"

- Line 61: "different regions"

Authors’ response: changes have been made as requested by the reviewer.

MATERIALS AND METHODS:

- Line 92: please specify what kind of antibodies (IgG I suppose) were detected by ELISA.

Authors’ response: IgG were detected by ELISA. Changes have been made as requested.

- Line 96: "and WNV antigens."

- Line 101: remove "virus"

- Line 102: "were equal"

Authors’ response: changes have been made as requested.

RESULTS:

- Line 151: Did you specifically detect IgG antibodies for DENV?

If you used the results of the HI assay to assess DENV seroprevalence (this is what I suppose based on the caption of Table 1 stating “Participants are considered positive for dengue if positive for at least one serotype”), I suggest removing "IgG".

Authors’ response: IgG antibodies were also detected by ELISA, then HI was conducted to determine titer of each serotype.

DISCUSSION:

- Line 298: "should not be excluded"

- Line 299: "that assesses"

- Line 301: "DENV1 serotype seemed"

- Line 303: "was responsible for"

- Line 306: "DENV4 has never been confirmed"

- Line 309: "West Nile virus"

Authors’ response: changes have been made as requested.

- Line 317: Did you specifically detect IgG antibodies for WNV?

Authors’ response: identically IgG antibodies were detected also by ELISA.

- Line 347: "some other flaviviruses"

- Lines 349-350: some seropositive individuals could also have acquired their infection in another country.

Authors’ response: changes have been made as requested.  However, we would like to mention that the survey was conducted among the Malagasy population with a high proportion living in rural areas. As Madagascar is an island, it is unlikely that individuals surveyed had travelled outside the country.

CONCLUSIONS:

- Lines 356-357: "an event of acute febrile illness outbreak"

- Line 358: "previous epidemics"

- Line 360: "DENV seemed"

- Line 364: "WNV transmission seemed"

Authors’ response: changes have been made as requested.

SUPPLEMENTARY MATERIALS:

Tables S1, S2 and S3: please indicate in the caption the signification of p-values shown in bold.

Authors’ response: changes have been made as requested.

Table S1: There is an error for the % of positive individuals among those aged 18-24 years (it should be 6.7% instead of 26.7%).

Authors’ response: many thanks for this remark, change has been made as requested.